# Global literature review and survey of implementation constraints on natural climate solutions

**Timm Kroeger** [1] ✉, **James T. Erbaugh** [1,2], **Zhixian Luo**[1], **Hilary Brumberg** [3,4,5], **Waverly Eichhorst**[3], **Margaret Hegwood** [3], **Anna LoPresti** [6], **Priya Shyamsundar** [1], **Peter W. Ellis** [7], **Lauren E. Oakes** [8,9], **Dow Martin**[10], **Pedro H. S. Brancalion** [11,12], **Mieke Bourne**[13], **Arundhati Jagadish** [14,15], **Kemen G. Austin** [9,20], **Andrew T. Kinzer**[16], **Marcos Sanjuán**[17,18], **Lisa McCullough**[15] & **Marta Echavarria**[19]

Natural Climate Solutions – protection, restoration, and improved management of lands and waters that reduce greenhouse gasses – have large climate change mitigation potential. However, lack of comprehensive information on implementation challenges hinders the adoption of Natural Climate Solutions and the delivery of near-term mitigation. Using a global survey of Natural Climate Solutions projects and a systematic review of recent studies, we map 46 constraints in eight categories, yielding 15,572 geo-referenced pathway-constraint observations from 501 studies and projects in 137 countries covering 20 of 22 United Nations subregions. Social-behavioral, Knowledge, and Government-Organizational are the most-reported constraint categories, and lack of policy coordination or implementation capacity the most-observed constraint and most frequently top-ranking constraint for pathways and subregions. Despite broad congruence, top constraint and category rankings vary among subregions and pathways, respectively. We find that projects generally encounter diverse sets of challenges. Without enabling efforts, near-term mitigation from Natural Climate Solutions may remain well below its biophysical potential.

Limiting global warming to less than 2 °C above 1850-1900 temperatures will require rapid and large-scale implementation of greenhouse gas (GHG) emission reductions and atmospheric carbon dioxide removal (CDR)[1–4]. Natural climate solutions (NCS)–deliberate human actions that protect, restore, and improve management of forests, wetlands, grasslands, oceans, and agricultural lands and that reduce GHG emissions or increase carbon dioxide sequestration, with no net negative impact on food and fiber supply or biodiversity and implemented in socially and culturally responsible ways[5–7]–are the least-cost mitigation option that could achieve large-scale, short-term CDR[1,8].

NCS are broadly equivalent to the Intergovernmental Panel on Climate Change (IPCC) supply-side measures related to agriculture, forestry, and other land uses (AFOLU)[1]. AFOLU mitigation measures have an estimated 2020-2050 mean total technical mitigation potential of 28.4 GtCO₂e yr⁻¹ (full range: 8.8-65.1[1]). In part due to their technical readiness[1,9] and cost-effectiveness, several NCS feature prominently in national climate change mitigation and adaptation strategies[10–13] and international mitigation scenarios[14–16], and their large-scale implementation is promoted by high-profile international initiatives[17,18], national governments (including via Nationally

**Fig. 1 | Number of studies and survey projects by country.** Colors indicate the total number of published articles and survey projects for each country that were included in the analysis. Basemaps are from Runfola, D. et al., geoBoundaries: A global database of political administrative boundaries, PLoS ONE 15(4): e0231866, https://doi.org/10.1371/journal.pone.0231866, published under a CC BY 4.0 Attribution 4.0 International license (https://creativecommons.org/licenses/by/4.0/).

Determined Contributions [NDCs] under the Paris Agreement) and private actors.

Yet NCS implementation remains limited relative to its technical potential[1]. Given the need for rapid and large-scale action during a climate-critical window[1,6,19–21] and the time required to achieve maximum NCS mitigation rates (e.g.[22]), such limited implementation is problematic. NCS implementation faces many challenges that vary in scale from the local (site or project) to the national level[23–26]. High-profile, global or pan-tropical analyses of NCS mitigation potential (e.g.[27]) often fail to consider the effect of local constraints on that potential[28,29]. Country-level analyses that consider even a select number of constraints find that the realistically feasible NCS implementation potential can be substantially below the biophysical or technological potential[30,31], supporting the IPCC's assessment that "[t]he economic and political feasibility of implementing AFOLU mitigation measures is hampered by persistent barriers"[1] (p.751).

Several recent global or pan-tropical analyses of NCS mitigation potential do consider implementation feasibility[23], constraints[26,32,33] or enabling conditions[24,33,34] of NCS. However, these studies focus on a small set of high-level indicators of national-level feasibility or enabling conditions[32,35]; report assessments of enabling conditions at the national level[24]; limit their analysis to a small, highly aggregated set of broad types of constraints[33]; or do not assess constraints spatially[26,33]. Thus, there is an urgent need for a geographically-comprehensive and more spatially-resolved mapping of constraints, at the finer spatial scales at which many constraints operate, and including a broader, more finely differentiated set of constraints that affect the feasibility of NCS implementation in different socio-ecological contexts[7,13,33]. Without this information, it is impossible to know how much of the mitigation potential estimated to be available at up to USD 100 tCO$_2$e$^{-1}$ may be achievable in the near to medium term.

Brumberg et al.'s[36] systematic review of the recent literature on NCS constraints was an important contribution to filling this critical knowledge gap. In this paper, we build on and advance that effort by conducting a global survey of NCS projects that uses an expanded set of implementation constraints, recoding Brumberg et al.'s[36] literature dataset to this expanded constraints set, georeferencing all survey and literature constraint observations to their corresponding administrative levels, and integrating the survey and literature datasets into a spatial database that reflects the current evidence base on NCS implementation constraints.

Our analysis advances and complements existing assessments in several ways. First, by focusing on reported constraints on NCS rather than enabling conditions or feasibility indicators of NCS[24,35], we identify which critical enabling conditions are currently absent, and where. Second, rather than report constraints, enabling factors or feasibility at the national level[24,32,35,36] or non-spatially (e.g.[26,33]), we collect and report constraints at the study or project level. Third, we distinguish constraints and discrete NCS mitigation options (hereafter, pathways[6]) in greater detail than previous studies[26,33,36]. Finally, our survey expands the existing evidence base on NCS constraints by including NCS projects not currently captured in the peer-reviewed literature. Our findings can help NCS planners anticipate challenges and inform efforts to identify solutions that can overcome constraints on NCS, enabling broader NCS uptake and effectiveness. A better, spatially-explicit understanding of the factors that constrain NCS adoption also can inform more realistic assessments of the feasible (vs the technical) mitigation potential[37,38].

## Results
### Sample frame
Our combined literature and survey dataset comprises 501 unique sources that provide spatially-explicit information on constraints to specific NCS pathways. This evidence base consists of 347 peer-reviewed studies from 133 countries (Figure S2) and 154 surveyed projects in 46 countries (Figure S3), with a combined coverage of 137 countries in 20 of 22 United Nations (UN) subregions (Fig. 1). The five countries with the highest number of papers or survey projects are Brazil (52), USA (49), Indonesia (32), China (28) and Kenya (26).

The Americas are the single most-studied region in our combined dataset with a total of 26 countries in the hemisphere covered a combined 243 times in papers and survey responses (Table S3). Africa and Asia follow with 179 counts each, covering 44 and 37 countries, respectively. Europe and Australia-New Zealand-Melanesia have 81 and 21 counts, respectively, covering 26 and four countries, respectively. Our sample covers 81% of all countries in Africa, 79% in the Americas, 77% in Asia, 59% in Europe, and 57% in Australia-New Zealand-Melanesia.

Africa accounts for a relatively high share of survey projects, with 63 projects (41% of all survey projects) in 22 countries (48% of all survey countries), followed by the Americas (59 projects [38%] in 15 countries [33%]), and Asia (25 projects [16%] in five countries [11%]) (Figure S3). In

the literature sample, the Americas are the most-studied region (184 country observations covering 25 countries), followed by Asia (154 observations, 37 countries), Africa (116 observations, 41 countries) and Europe (79 observations, 26 countries). Figure S4 shows the spatial coverage of our combined sample by administrative level. Forty percent of our literature sample are country-level (ADM-0) studies, with the remainder approximately evenly split between state-level (ADM-1) and county-level (ADM-2). By contrast, 60% of the survey projects most closely represent ADM-1 level interventions in terms of their implementation extent, followed by ADM-2 (27%) and ADM-0 level (13%).

In 29 cases, a group of individuals completed a survey for a project, resulting in a total of 215 survey participants from 40 organizations representing environment (38%) and development (18%) NGOs, research institutions (18%), governmental entities including utilities (10%), multi-sectoral entities (10%) and for-profit enterprises (8%).

## NCS pathway representation

The number of papers in our sample (347) exceeds that of surveyed projects (154), with papers contributing the majority (60%) of the overall pathway observation count (i.e., the number of times a pathway is reported in papers and survey responses) as well as the counts for most pathways (Figure S5). Yet, on average, surveyed projects implement more pathways (2.5) than are reported on in the literature sample (1.5). In both the literature and survey samples, reforestation is the pathway with the largest share of observations (27% and 25%, respectively), followed by agroforestry (25% and 14%, respectively) and avoided forest conversion (20% and 14%, respectively). Reforestation is the most-represented pathway in our data, with a combined 220 papers and survey projects (44% of all papers or survey projects), followed by agroforestry (172; 35%) and avoided forest conversion (147; 30%) (Figure S5). Regenerative agriculture, which was not included in Brumberg et al.[36], accounts for only five percent of observations. Fewer than 20 papers or projects report on peatland restoration, avoided peatland conversion, avoided grassland conversion, and reduced woodfuel harvests in forests, respectively. Given the dominance of reforestation, agroforestry and avoided forest conversion, it is not surprising that restoration is the most-represented NCS strategy (277 papers and survey projects; 56%), followed by improved management (249; 50%) and protection (176; 36%).

Eastern Africa and South America contribute by far the largest numbers of papers and survey responses at the pathway level (155 and 150, respectively), followed by South-Eastern Asia (86), Northern America (83), Central America (73) and Western Africa (72) (Figure S6). Central Asia (2), Southern Africa (7), and Melanesia, Eastern Europe and Western Asia (8 each) have the lowest numbers of papers and survey responses. In most of the 20 global subregions, reforestation or avoided forest conversion are the single most-represented pathways in the combined literature and survey sample. The exceptions are Western and Eastern Africa and Southern Asia, where agroforestry is the most-represented pathway.

## NCS implementation constraints

Our combined dataset contains 15,572 constraint observations, each of which identifies a specific implementation constraint for a specific NCS pathway in a specific location. Approximately three quarters (73.5%; n = 11,448) of observations are from the survey, despite the survey accounting for only one-third of the combined number of survey responses and papers. This inverse relationship is due to surveyed projects reporting higher numbers of implemented pathways and constraints on average than the average paper in the literature review. This evidence base is distributed highly unevenly across pathways (Figure S7) and UN subregions (Figure S8), and pathway-subregion combinations (Figure S9).

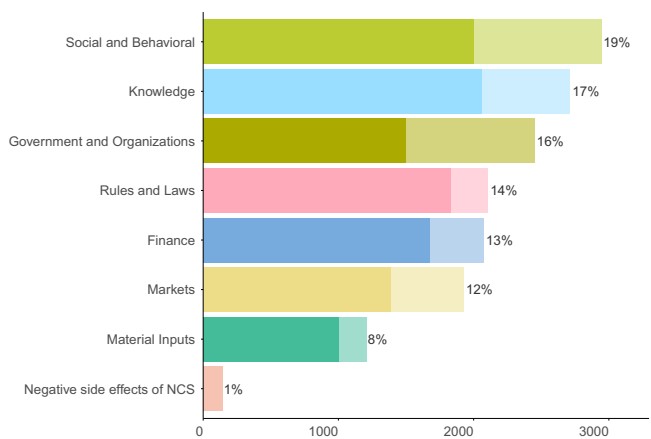

**Fig. 2 | Frequency distribution of constraint observations by category.** Solid and partially transparent bar sections show observations contributed by the survey and the literature, respectively. Percentages next to bars show the share of each category in total constraint observations.

## Most-reported constraint categories overall

There are no constraint categories whose observation counts dominate (Fig. 2). Even the sixth-most reported category (Markets) still has two-thirds of the observation count of the most-reported (Social-behavioral). Social-behavioral constraints (e.g., skepticism or disinterest in NCS or lack of trust in NCS promotors, lack of social learning or exchange networks, preferences for non-NCS, lack of coordination among interested parties, concerns about equity) was the most frequently reported constraint category in the combined sample, accounting for nearly one-fifth (19%) of all constraint observations, followed by Knowledge (17%), Government and Organizations (16%), Rules and Laws (14%), Finance (13%), Markets (12%) and Material Inputs (8%), with Negative Side Effects by far the least-observed constraint category (1%) (Fig. 2). In the literature sample, Government and Organizations was the most-observed constraint category, followed closely by Social-behavioral, and then by Knowledge (Figure S11). In the survey sample, Knowledge is the most-reported category, followed closely by both Social-behavioral and Rules and Laws, respectively. The survey data show a more even distribution of observations among constraint categories than the literature (Figure S11).

## Most-reported individual constraints and high prevalence of many constraints

Our results show that many constraints identified by individual NCS papers or projects are encountered frequently in NCS implementation. In fact, except for the most-reported constraint, there is no clear break in the frequency count distribution of our 46 constraints (Fig. 3). Similarly, the well-mixed constraint categories (indicated by the different bar colors in Fig. 3) show that constraints in all categories except Negative Side Effects are encountered often, with the top five most-reported constraints representing four different constraint categories. Both the generally smooth frequency count distribution across individual constraints and the high degree of mixing of constraint categories in Fig. 3 show that the overall frequency ranking of constraint categories (Fig. 2) is driven by a large number of often-encountered constraints, not a small number of dominant constraints.

The most frequently reported constraint was lack of policy coordination or implementation capacity, followed by uncertain, or lack of, enforcement of environmental laws (Fig. 3), both of which fall in the Government and Organizations category. Lack of information about how to design the NCS, skepticism or disinterest in NCS or lack of trust in NCS promoters, and project access to other funding for NCS round out the top-five constraints, followed closely by lack of non-credit funding for land managers.

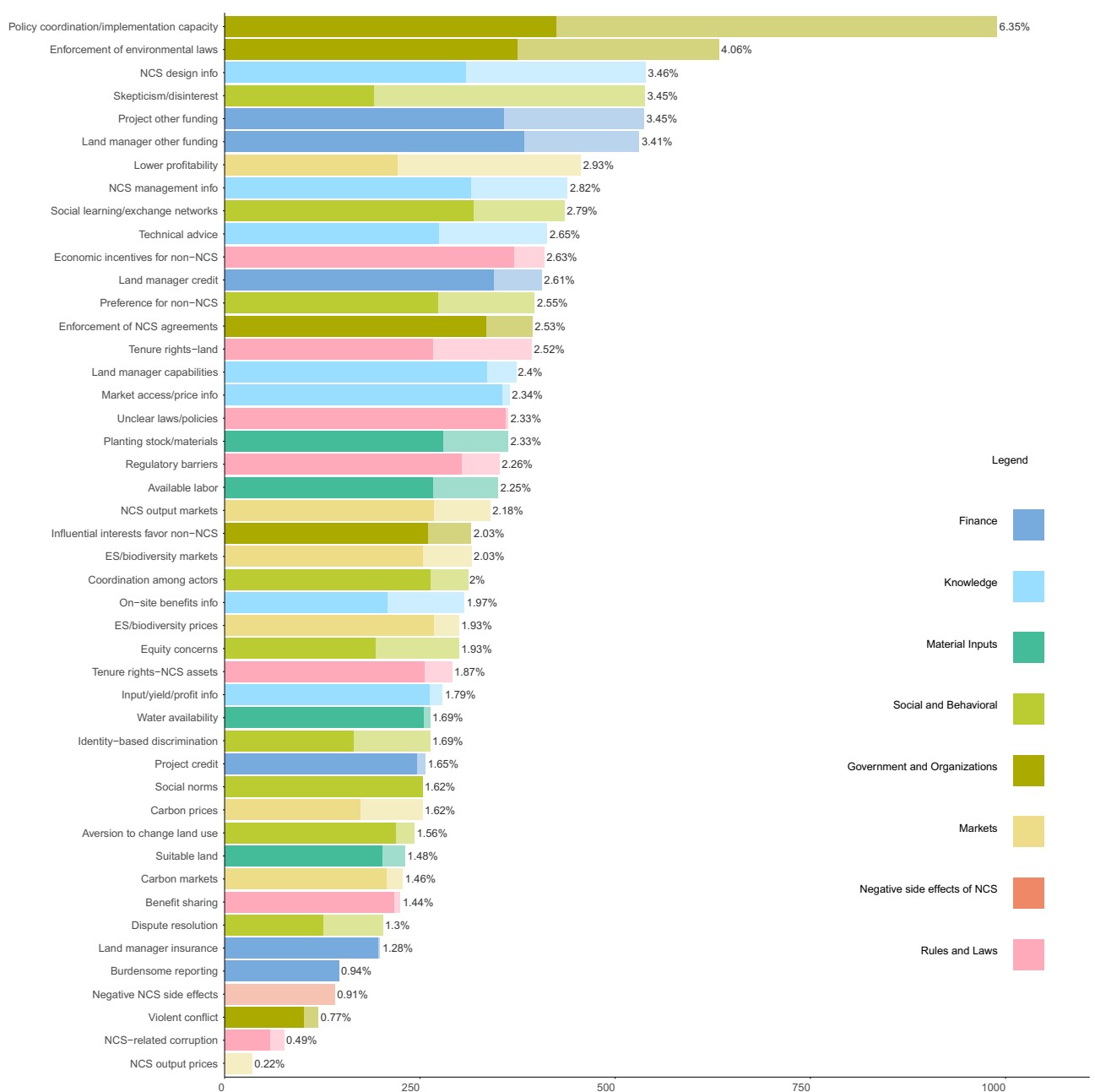

**Fig. 3 | Frequency counts of individual constraints.** Solid bar sections show survey observations; partially transparent sections, literature observations. Abbreviated constraint labels shown. See Table S1 for full constraint labels. Percentages next to bars show the share of each constraint in total constraint observations.

Lack of policy coordination or implementation capacity also was the most-frequently reported constraint in the individual literature and survey samples. In the literature data, this was followed by skepticism or disinterest in NCS or lack of trust in NCS promoters; uncertain, or lack of, enforcement of environmental laws; greater profitability of alternative land uses; and information about how to design or begin the NCS. In the survey data, the second-most-frequent constraint was land manager access to other funding for NCS, followed by uncertain, or lack of, enforcement of environmental laws; unclear laws and policies related to NCS outputs/markets; and financial or other incentives for non-NCS.

### NCS projects face multiple, diverse constraints

Implementation generally encountered multiple constraints in multiple categories, with an overall weighted average of 7.8 constraints in 3.2 categories per paper or project. This is true for both the literature

(average: 2.7 constraints in 2.2 categories per paper) and survey samples (average: 19.3 constraints in 5.6 categories per project), but especially the latter, which is not surprising since the survey explicitly aimed to identify the full range of constraints, while papers frequently tested for or analyzed specific constraints or did not have the identification of constraints as their main research objective. These findings support the observation that NCS implementation commonly encounters diverse challenges, facing a complex set of constraints that require integrated solutions[39].

### Relative prevalence of constraint category observations by pathway

Given the uneven distribution of the constraints evidence base across pathways (Figure S12; Figure S7), to facilitate comparisons of the relative prevalence of constraint categories among pathways, Fig. 4 shows a z-score heatmap of the number of observations in each

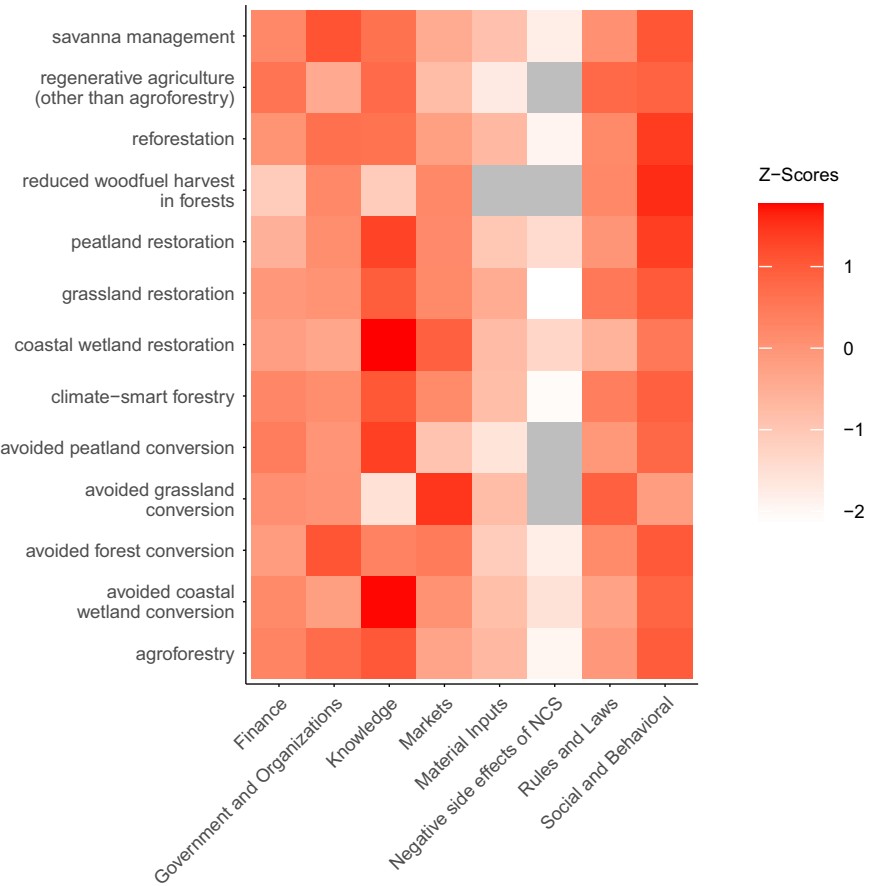

**Fig. 4 | z-score heatmap showing the relative frequency with which each constraint category was reported for a given NCS pathway.** Z-scores calculated at the pathway level. The color in each cell indicates the number of standard deviations that the number of observations of the respective constraint category is above the mean of observations for all constraint categories for that pathway, where darker shading indicates more standard deviations above the mean, and lighter shading more standard deviations below the mean. For example, for both the avoided coastal wetland conversion and the coastal wetland restoration pathways, Knowledge constraints have a z-score of 1.8, the highest score of any constraint category for these two pathways, indicating that Knowledge constraints are reported with a frequency that is nearly two standard deviations above the mean of all constraint categories reported for these pathways. Gray indicates no data.

constraint category normalized at the pathway level. Social-behavioral and Knowledge are the top-ranking constraint categories for the highest number of pathways (five each), followed by Government and Organizations (two) and Markets (one). Social-behavioral also is the second-highest-ranking constraint category for five additional pathways, and the category with the highest combined top-three rankings among pathways (Figure S13).

Social-behavioral also is the most-reported constraint category for all three NCS strategies (Protection, Restoration, Improved Management), with Knowledge and Government and Organizations the second and third-most reported categories depending on strategy (Figure S14).

### Relative prevalence of constraint category observations by subregion

Social-behavioral is the most frequently number one-ranking constraint category at the subregion level (8 of 20 subregions), followed by Markets (6 subregions) and Knowledge (5 subregions) (Fig. 5). Social-behavioral, Knowledge, Markets, and Government and Organizations are the constraint categories with the highest numbers of top-three rankings in subregions (17, 14, 8, and 8, respectively) (Figure S15, right panel). The approach used to aggregate constraint category observations across pathways in each subregion (summing raw counts vs summing pathway-level z-scores) only marginally affects these results: Social-behavioral is the most-frequently number one-ranking constraint category under both approaches, and

Social-behavioral, Knowledge, and Market constraints are the most-frequently top-three-ranking categories, although Knowledge displaces Markets as the second-most frequently top-ranking category under pathway-level z-score aggregation (Figure S15). Material Inputs consistently is the constraint category with the fewest top-three rankings overall at the subregion level (0 or 1), followed by Negative Side Effects and Rules and Laws (3 times each). Dropping subregions with fewer than 30 or 50 observations does not affect these rankings.

### Congruence of most-reported constraint categories among NCS pathways and UN subregions

A key finding is the broad consistency of the most prevalent constraint categories among pathways and subregions, respectively.

While their specific rankings may vary among pathways and among subregions, Social-behavioral, Knowledge, Markets, and Government and Organizations constraints are highly prevalent for nearly all pathways and subregions, while Finance and Rules and Laws are important for large subsets of pathways and subregions as well. Notably, Finance is a top-three-ranking constraint category in seven—one-third of all—subregions (based on raw counts; five under pathway-level z-score aggregation [Figure S15]), including in all five African subregions. Finance also is the third-most frequently-reported constraint category for three pathways, all of which are avoided conversion pathways (i.e., protection). The latter is perhaps not surprising given that actual avoided conversion results in the foregoing of

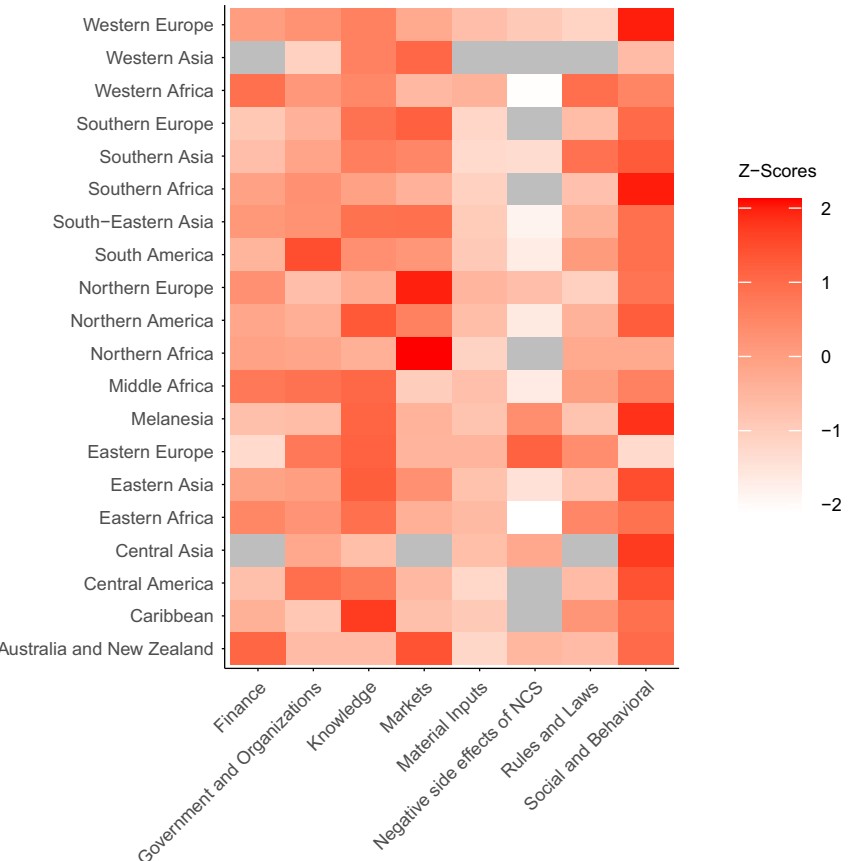

**Fig. 5 | z-score heatmap showing the relative frequency with which each constraint category was reported for a given subregion.** Z-scores calculated at the subregion level. The color in each cell indicates the number of standard deviations that the number of observations of the respective constraint category is above the mean of observations for all constraint categories for that subregion, where darker shading indicates more standard deviations above the mean, and lighter shading more standard deviations below the mean. Gray indicates no data.

prospective private benefits derived from the alternative land uses (e.g.[40]), which, at least on private or community-owned lands, but also on multi-use public lands or public lands lacking effective enforcement of property rights, often requires financial incentives[41–46].

## Most-reported individual constraints by pathway

Five (out of 46) constraints are the highest-ranking constraints for all 13 pathways (Figure S18). If the reduced woodfuel harvest in forests pathway, which has only five papers or survey responses, is excluded, only three constraints are top-ranking for all pathways. Lack of policy coordination or implementation capacity was the most-observed constraint for eight (62%) of the 13 pathways, and ranking second or third, respectively, for two addition pathways. Information about how to design or begin the NCS was the most-frequently reported constraint for four pathways (avoided coastal wetland conversion, coastal wetland restoration, avoided peatland conversion, and peatland restoration, for which it was tied with lack of policy coordination or implementation capacity), while greater profitability of alternative land uses was the most frequently reported constraint for one pathway (avoided grassland conversion). The constraint frequency distributions for all pathways are fairly smooth overall (Figure S19). However, eight pathways (62%) show a marked discontinuity, with the top one or two constraints much more frequently reported than the next constraints. This is true for agroforestry (lack of policy coordination or implementation capacity), avoided coastal wetland conversion (information about how to design or begin the NCS), avoided forest conversion (lack of policy coordination or implementation capacity and uncertain, or lack of, enforcement of environmental laws), climate-

smart forestry (lack of policy coordination or implementation capacity and information about how to manage the NCS), coastal wetland restoration (information about how to design or begin the NCS), reforestation, and savanna management (lack of policy coordination or implementation capacity).

Interestingly, lack of planting stock, while often mentioned as a key constraint in reforestation analyses (e.g.[47,48]), does not emerge as a dominant constraint in our data, accounting for only two percent of all constraint observations in the literature and survey data, respectively. For reforestation, lack of all material inputs (planting stock or other materials; labor [external or own]; land; or water for the NCS) combined accounts for only eight percent of all constraint observations, close to the pathway mean (7%) (Figure S18), while lack of planting stock or other materials itself is reported only 114 times by 220 papers and projects that reported on this pathway, ranking it twelfth out of 46 reforestation constraints in terms of frequency count. This does not mean that seedlings and other inputs are not an important constraint on large-scale reforestation efforts or in specific geographies (e.g.[49]), rather that they are not an apparent dominant constraint across most reforestation projects, or across NCS pathways generally. When other constraints are overcome, reforestation initiatives are able to organize the seedling supply[50].

## Most-reported individual constraints by subregion

Twelve constraints are highest-ranking in at least one of the 20 subregions (Figure S20), or ten if subregions with <10 studies and survey responses are excluded. Lack of policy coordination or implementation capacity is the top constraint in seven subregions (35%) and the

second or third-most-reported constraint for a further five. Skepticism or disinterest in NCS or lack of trust in NCS promoters and markets for ecosystem services and biodiversity provided by the NCS are the most-reported constraints for the next-highest number of subregions (three each). Land manager access to other funding for NCS is tied with skepticism or disinterest in NCS or lack of trust in NCS promoters and markets for ecosystem services or biodiversity provided by the NCS for the second-highest number of combined first and second-place rankings, and has the second-highest number of top-three rankings, after lack of policy coordination or implementation capacity. Subregions where lack of policy coordination or implementation capacity is not the most-frequently reported constraint are Central Africa, where uncertain, or lack of, enforcement of environmental laws is the most-reported constraint, which is also second-ranking in both South America and Central America; Australia and New Zealand and Eastern Europe (small n), for which negative side effects of NCS is the top constraint; Southern Europe, Northern Africa, and Western Asia (small n) (markets for ecosystem services or biodiversity provided by the NCS); the Caribbean (information about how to design or begin the NCS, land manager access to other funding for NCS); Northern Europe (greater profitability of alternative land uses); and Western Europe and Eastern Asia (skepticism or disinterest in NCS or lack of trust in NCS promoters). As was true for some pathways, several subregions show exponential or quasi-exponential frequency distributions with one or a few constraints with much higher counts than the next most frequent constraints (Figure S21). This is true for Northern, Central, and South America and Southern Asia (lack of policy coordination or implementation capacity; and in South America, also uncertain, or lack of, enforcement of environmental laws); Eastern Asia (skepticism or disinterest in NCS or lack of trust in NCS promoters); Australia and New Zealand (negative side effects of NCS); Northern Africa (markets for ecosystem services or biodiversity provided by the NCS, markets for NCS outputs, and greater profitability of alternative land uses); Northern Europe (greater profitability of alternative land uses); Southern Europe (markets for ecosystem services or biodiversity provided by the NCS; skepticism or disinterest in NCS or lack of trust in NCS promoters, and availability of technical advice for land managers); and Western Europe (skepticism or disinterest in NCS or lack of trust in NCS promoters and availability of technical advice for land managers).

**Congruence of most-reported constraints among NCS pathways and, to a lesser extent, UN subregions**

The top individual constraints highlight the diverse nature of the key implementation challenges faced by many NCS projects. Lack of policy coordination or implementation capacity is the single most frequently reported individual constraint in both the survey and literature samples, followed by lack of enforcement of environmental laws, highlighting the importance of lack of institutional capacity as a key barrier[32,35]. The next four constraints have nearly identical frequency counts, indicating a widespread lack of NCS design information, skepticism or disinterest in NCS or lack of trust in NCS promoters, and lack of funding for NCS on the part of both land managers and projects. Indeed, if the two funding constraints (land manager; project-level) were combined, funding would be the most-reported individual constraint overall, just edging out lack of policy coordination or implementation capacity —not surprising given the current large funding gap for NCS and CDR more broadly[51], and confirming the general view of funding as a key constraint for NbS more broadly[52,53]. Together, these six constraints account for one quarter of all observations for the 46 constraints.

The top-ranking overall constraints are not driven primarily by a small set of pathways or subregions with high observation counts in our sample, but rather by high rankings of these constraints for most pathways and subregions. Lack of policy coordination or implementation capacity, the top overall constraint, also is the top-reported

constraint for eight of 13 pathways (62 percent) and seven of 20 subregions (35 percent) and is a top-five constraint for ten pathways and 15 subregions. Information about how to design or begin the NCS (third-ranking overall) is top-ranking and top-five-ranking for four and six pathways, respectively, and one and seven subregions, respectively. Skepticism or disinterest in NCS or lack of trust in NCS promoters, the fourth-ranking constraint overall, is second-ranking for two and top-five-ranking for five pathways, and top-ranking in three and top-five-ranking in nine subregions, respectively. Land manager access to other funding for NCS (sixth-ranking overall) is top-ranking in two, and top-five-ranking in nine subregions, respectively; and top five-ranking for five pathways. There are exceptions to this pattern of high overall rankings of individual constraints being driven by their high rankings for many pathways and subregions. The second-place overall ranking of uncertain, or lack of, enforcement of environmental laws results from its high observation counts in the large-sample South and Central America and Western and Eastern Africa subregions and its high ranking (second or third place) in the three pathways with the highest numbers of constraint observations (reforestation, agroforestry, avoided forest conversion), despite being the top-ranking and top-five-ranking constraint in only three and five subregions, respectively. Similarly, the fifth-place overall ranking of project access to other funding for NCS —which is not top-ranking for any pathway or subregion— is driven predominantly by its relatively high (fifth-place) ranking in the three subregions with the highest observation counts (Western Africa, Eastern Africa, South America) and its sixth-place ranking in the subregion with the fourth-highest observation count (Central America).

Despite broad congruence among subregions and pathways in terms of leading constraint categories and constraints, some key differences emerge. At the broad regional to continental level, in Africa, Knowledge, Social-behavioral and Finance constraints are most prevalent; in East and South Asia, Social-behavioral constraints rank highest, followed by Knowledge and Markets constraints. In Central and South America, Social-behavioral and Government and Organizations constraints are most prevalent, followed by Knowledge constraints; while in Europe, Social-behavioral constraints are most prevalent, followed by Markets and Knowledge constraints.

Uncertain, or lack of, enforcement of environmental laws is the top constraint in Central Africa, and second-ranking in South and Central America. Lack of funding, while an important barrier in most subregions, is particularly prevalent in Africa: in all African subregions except Northern Africa, either land manager access to credit for NCS or land manager access to other funding for NCS is the most or second-most reported constraint, with the respective other constraint also ranking high. Moreover, project access to funding is the fifth-most frequently reported constraint in Eastern, Western, and Central Africa. Limited social learning networks, land manager literacy, numeracy or technological capabilities, concerns over negative equity impacts, and politically influential interests favoring non-NCS all are barriers particularly prevalent in Africa, as evidenced by the relatively darker shading of these constraints for these sub-regions in Figure S20 (note that the figure is color-coded at the subregion level, so the shading is comparable across subregions as an indicator of the relative prevalence of a constraint across all subregions).

Greater profitability of alternative land uses is the most-reported market constraint overall, but in some subregions, other market constraints rank higher: in Southern Asia and Eastern Africa, carbon prices, markets for NCS outputs (i.e., food and fiber), and prices for ecosystem services or biodiversity all rank higher; in Australia and New Zealand, carbon prices, and markets and prices for ecosystem services or biodiversity rank higher. This points to the importance of missing or weak markets for NCS products, ecosystem services or biodiversity provided in large portions of these subregions, caused in part by high transaction costs[54], the public good nature of some of these outputs[55],

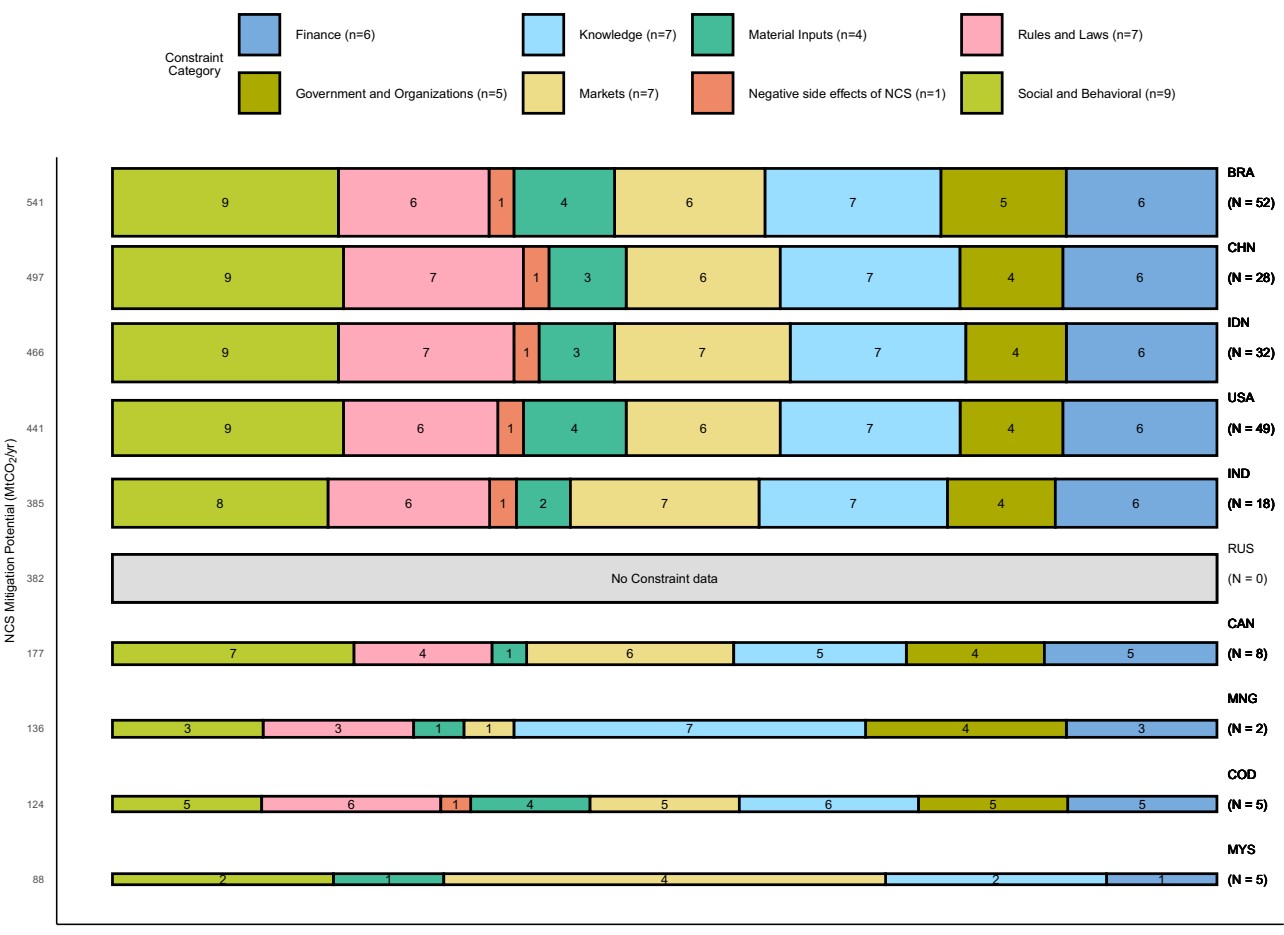

**Fig. 6 | Composition of NCS constraints in ten countries with highest NCS biophysical mitigation potential.** Bar segments represent constraint categories, with segment length proportional to a constraint category's share in total constraint observations for a country. Numbers in segments indicate number of different constraints in a category recorded in our database for that country. Numbers in legend (n) indicate number of constraints tracked in each category in this study. Numbers on vertical axis show estimated annual NCS biophysical mitigation potential of each country as reported in[166], used to scale bar height. *N* indicates the number of papers and projects for a country in our database. BRA, Brazil; CHN, China; IDN, Indonesia; USA, United States of America; IND, India; RUS, Russian Federation; CAN, Canada; MNG, Mongolia; COD, Democratic Republic of the Congo; MYS, Malaysia.

or lack of accounting frameworks that capture the full social value of natural assets and their services[56,57].

These differences suggest that efforts to enable broader implementation of specific pathways often may require tailored bundles of interventions in different subregions.

Among the large-sample subregions, Western and Eastern Africa stand out by their much thicker (fat-tailed) frequency distributions and the absence of one or two dominant constraints (Figure S21). These two subregions thus are characterized by a comparatively larger number of highly prevalent constraints on NCS implementation. In contrast, Northern America, Eastern and Southern Asia, and especially South and Central America and South-Eastern Asia with their near-exponential frequency distributions are characterized by a smaller number of much more prevalent constraints. All else equal, this suggests that enabling broad NCS uptake may require addressing a larger number of barriers across larger geographies in Western and Eastern Africa than in the other large-sample subregions.

Top-ranking constraints are more similar among pathways than among UN subregions. There are only five top-ranking constraints across 13 pathways —three if the reduced woodfuel harvest in forests pathway, which has only five papers or survey responses, is excluded. In contrast, there are 12 top-ranking constraints across 20 subregions. For example, lack of policy coordination and implementation capacity is the most-frequently top-ranking constraint for both pathways and

subregions, but it is so for only 35 percent of subregions versus for 62 percent of pathways. Even adjusted for the greater number of subregions compared to pathways, diversity in top constraints is over 50 percent higher among subregions than pathways. This is due to differences among regions in both the relative size of the evidence base for different pathways, and the ranking of individual constraints (Figure S21) and constraint categories for given pathways (Figure S17). As was true for subregions, the somewhat thicker constraint frequency distributions exhibited by some large-sample pathways (regenerative agriculture, grassland restoration, and climate-smart forestry) indicate that projects implementing these pathways on average face more constraints.

## Constraint composition in ten countries with highest NCS biophysical mitigation potential

The countries with the largest identified NCS biophysical mitigation potential show a surprising similarity in the relative prevalence of constraint categories, represented by the colored bar segments in Fig. 6. As in our overall sample (Fig. 2), in each of the top-five countries by biophysical mitigation potential, Social-behavioral constraints account for the highest, and Knowledge generally the second-highest share of constraint observations, though in China and India, the latter are tied with Rules and Laws. Government and Organizations constraints account for a notably smaller share (9-11%) of total constraint

observations in each of the top five mitigation potential countries than they do in our full sample (16%), while Rules and Laws and Markets account for slightly larger shares than in the full sample. Importantly, despite the limited number of papers and surveyed projects in each country (N ≤ 52), nearly all (41-44) of the 46 tracked constraints are reported in each country with at least ten papers or survey projects.

## Discussion

We combine a systematic review of the recent peer-reviewed literature on NCS barriers with a global survey of projects that implement NCS to analyze a set of 46 constraints that encompasses the full range of NCS implementation barriers, from material inputs to market, finance, knowledge, social-behavioral, legal-regulatory, and governmental and organizational challenges. Using information from 347 papers and from 154 projects not represented in the literature, we construct the – to the best of our knowledge – largest, geographically broadest, most comprehensive and detailed (in terms of pathway coverage and breadth of included constraints), geo-referenced evidence base on NCS implementation constraints to date. The literature and survey samples have different regional and pathway distributions, with the survey showing a somewhat more even pathway representation and expanding country coverage in Africa and the Caribbean. The two data sets also yield different relative prevalence rankings of the constraint categories but coincide in the most prevalent categories. By probing comprehensively for the presence of constraints at the project level, the survey data from 46 countries more than triple the size of the evidence base on NCS implementation constraints in the recent peer-reviewed literature.

Four key insights emerge from our findings. NCS implementation faces a large number of highly prevalent constraints. These constraints are diverse in type, with the most prevalent constraints representing all constraint categories tracked in this study except Negative Side Effects of NCS and Material Inputs. Moreover, NCS projects generally face multiple constraints in different categories. Finally, the most-reported constraint categories are broadly consistent among NCS pathways and UN subregions, respectively, despite some notable variation.

In the following, we discuss the role of financing and the importance of targeting in removing NCS implementation constraints. We also provide guidance on targeting interventions to resolve constraints and on how to enable broader NCS uptake, and discuss the limitations of our analysis.

Our findings reveal that while available financing certainly constrains NCS implementation, finance-related constraints neither individually nor collectively are the most-reported constraints for most pathways or UN subregions. Yet this does not mean that NCS implementation is not funding-constrained. For one, investment in NCS and the AFOLU sector in general has been much lower than what is needed to achieve the Paris Agreement climate targets[1,58] and is geographically misaligned with NCS opportunities, with upper-middle, lower-middle, and low income countries (other than China) jointly accounting for 90 percent of the investment opportunity in nature protection or restoration during 2020-30 but only 20 percent of the investment[59]. Importantly, existing assessments of NCS finance needs rarely take into account the full cost of creating the missing enabling conditions for NCS projects – that is, of overcoming the social-behavioral, knowledge, market, legal and regulatory, and government and organizational constraints that our analysis shows are widely prevalent. Indeed, the cost of implementing critical enabling actions such as stakeholder engagement, extension services, social learning and exchange networks, improved smallholder credit or market access, or legal or policy reforms often are poorly understood or thought of as transaction costs from a narrow NCS implementation perspective and ignored in cost analyzes, even though they frequently account for a substantial share of total project costs (e.g.[60]). While additional funding by itself may not be sufficient to overcome these constraints, it

certainly will be necessary to finance enabling actions. These non-financial challenges are not unique to NCS. As Matthews and Wynes[61] point out, social inertia and technological and political factors, which broadly correspond to our three most-observed constraint categories, are crucial barriers to climate action on the needed scale for all mitigation approaches.

Our finding of near-ubiquitous high prevalence of multiple constraint categories for nearly all pathways and subregions indicates that unlocking additional large-scale NCS potential will require multi-pronged, integrated efforts that tackle multiple diverse barriers. This makes it imperative that interventions be targeted based on assessments of the causal relationships among constraints. Doing so can inform the design of more efficient interventions that focus on ultimate rather than proximate constraints. For example, consider some of the most-reported constraints in our dataset. Three of the top ten overall constraints are information-related (information about how to design or begin the NCS; information about how to manage the NCS; availability of technical advice for land managers), underlining the role of insufficient information among potential adopters as a key obstacle to NCS adoption. Skepticism or disinterest in NCS or lack of trust in NCS promoters is the fourth most-frequently reported constraint, and greater profitability of alternative land uses the seventh-most. However, lack of information itself can be a driver of disinterest or skepticism in NCS[62–64] or of their perceived lack of competitiveness. For example, Gladkikh et al.[65] found that some coffee growers in their sample erroneously thought that only conventional, full-sun coffee farming was eligible to receive state agricultural incentives, thus negatively impacting the perceived competitiveness of shade-grown coffee. Similarly, to the extent that skepticism or disinterest are caused by lack of information about NCS options or their performance or by suspected ulterior motives of NCS promoters, these constraints could be overcome by peer-to-peer social learning networks, demonstration projects, extension services, or information provided by other trusted sources. However, with lack of social learning or exchange networks and lack of technical advice themselves being top-ten constraints in our dataset, it is clear that neither is sufficiently available to many prospective NCS adopters. Furthermore, while scientific understanding of NCS climate mitigation performance is advancing[9], substantial uncertainty remains in many cases about their site-specific performance on other decision-relevant outputs (e.g., disaster risk reduction, crop yields, water security) or their financial competitiveness with alternatives (e.g., farm profit or cash flow; life-cycle cost). While the relevant evidence base is improving rapidly[66–73], this evidence is not necessarily readily incorporated into practices. For example, consultants and engineering companies often lack NCS expertise (e.g.[74,75]).

Similarly, availability of funding for NCS projects and availability of funding for land managers are leading constraints on NCS in our dataset (ranking fifth and sixth overall, respectively, and higher in several subregions), while regulatory barriers or unclear laws and policies related to NCS, lack of tenure rights, and politically influential interests that favor non-NCS all are well-represented in our data. Yet, perceptions of operational, political and regulatory risks themselves strongly condition available NCS funding[39]. Where this is the case, efforts to increase funding for NCS may be best directed at addressing the relevant regulatory, legal and policy constraints, or funders' perception of these constraints.

The recognition that constraints often may be interrelated means that there is reason for hope but also need for realistic expectations. On the hopeful side, a focus on a well-chosen set of ultimate (vs proximate) constraints can remove or mitigate additional constraints and be more efficient and feasible to implement than attempts to separately address the full suite of constraints. Furthermore, individual interventions that can address specific common barriers for individual NCS pathways generally are well-identified (e.g.[26]).

Yet which constraints interventions should prioritize requires an in-depth understanding of the local context to allow distinguishing ultimate from proximate barriers to NCS implementation. The effort needed to rigorously diagnose ultimate constraints is well-justified, however: our subregional constraints mapping by NCS pathway (Figure S17) shows that many constraints affect multiple or all pathways in a subregion. Removing or mitigating a particular ultimate constraint thus may not only mitigate related proximate constraints for the target pathway but likely also for additional pathways.

Our finding that Social-behavioral, Knowledge, and Government and Organizations are the most-reported constraint categories with regulatory and legal constraints also widespread further cautions against unrealistic expectations. It suggests that addressing economic (e.g., lower profitability of NCS, lack of (access to) markets for NCS outputs, or low prices for those outputs) and financial constraints (e.g., lack of access to credit or other funding for NCS, or lack of insurance for NCS), while undoubtedly important and often crucial[52], by itself is unlikely to enable a step-change in NCS adoption.

Enabling such a step-change demands an integrated, holistic, systems approach that removes multiple barriers, rather than piecemeal efforts to mitigate individual constraints[76–78]. Depending on the specific context, such integrated solutions may require the following:

Effective empowerment of Indigenous Peoples and local communities (IPLC) and smallholder farmers to adopt NCS. Indigenous Peoples manage or have tenure rights over one quarter of all lands globally[79], and smallholders (including Indigenous people) manage three-quarters of all agricultural lands globally[32]. Thus, IPLC and smallholders play crucial roles in any efforts to scale up NCS adoption and resulting climate change mitigation[80,81]. Effective empowerment of IPLC and smallholders requires secure and complete tenure rights[82], recognition of IPLC as leaders of NbS[83], access to information about NCS costs and benefits in locally appropriate forms, and equitable access to private and government services associated with NCS adoption and land management more broadly, including credit, insurance, financial incentive or subsidy programs, and extension services;

Community ownership[39] and co-design[7] of NCS interventions, to build trust and ensure NCS projects meet local needs and reflect local preferences and aspirations[83–87];

Identification and communication of likely local benefits NCS may provide, which often are the primary drivers of adoption (e.g.[88,89]). Comprehensive assessments of the evidence base on NCS non-climate mitigation benefits ('co-benefits')[90–94] can inform those efforts, but validation with local stakeholders of potential NCS benefits and their expected timing and identification of potential trade-offs with development goals remain essential[94–96];

Equitable distribution of societal co-benefits[97] and costs[98] of NCS. This can be achieved through private or public financial compensation schemes (e.g., PES or subsidies) for specific positive externalities (uncompensated off-site benefits received by third parties) of NCS;

Removal of cultural, gender and other discrimination to participation in NCS and active integration of gender equality and social inclusion in NCS programs[99–102], by identifying groups at risk of exclusion and their respective inclusion gaps, designing projects such that they close these gaps, and monitoring the inclusivity of impacts with the help of appropriate indicators[103,104];

Strengthening of the technical and organizational capacity of both public and non-governmental actors[23], particularly at the local and meso levels[105]. Such capacity improvements are necessary to allow better intra and interinstitutional coordination of NCS actions and related activities (e.g., agricultural extension, smallholder value chain integration); enable these institutions to provide decision-relevant information on site-level resource management (e.g., for tree-site matching, the WhichTreeWhere? guide[106]) or the Africa Tree Finder[107];

for weather and climate-adaptive crop management, the Climate Information for Grains tool[108]);

Development of social learning and exchange networks such as communities of practice to facilitate peer-learning[109–112] for NCS design and management, supported by technical assistance and extension services[113] and demonstration and experimentation sites that facilitate knowledge co-creation[105,114];

Creation or strengthening of supportive policy and legal frameworks for NCS[115], through harmonization of conflicting laws and regulations[39,52] and improved monitoring and enforcement of compliance with environmental laws[116,117] and agreements more broadly[118]. Such measures also reduce political and regulatory risk for NCS funders, which constrain funding flows[39];

Creation of a local economic base for NCS, through NCS-related enterprise development and business extension services[59,82,105], improved market integration of and value capture by smallholders for both NCS-related and other outputs, through deployment of socially inclusive information and communication technologies[119], producer cooperatives, outgrower schemes, offtake agreements, and other forms of value chain integration[120–124], navigating the challenges in creating inclusive agricultural value chains[125,126]; and

Mobilization of increased and more inclusive financing for NCS from both traditional and non-traditional sources, in part through innovative approaches such as facilitating agricultural value chain finance[127], using public finance to leverage private investment[128], blending development and private finance to de-risk private investments[52,129], PES[130–133] and redirecting of harmful government subsidies[83,134,135]. In addition to increasing NCS financing, access to financing must become more equitable[102,136–138]; and

Investment in research on NCS GHG mitigation effects in ecosystems whose GHG flux dynamics are less well understood, such as some peatlands and coastal wetlands[9], to reduce uncertainties about the size of the climate mitigation outcomes of specific intervention designs.

Because many of the interventions needed to remove NCS constraints seek to change human behaviors, such efforts should heed the principles of human-centered design-based approaches, which have been shown to improve outcomes of health interventions[139,140] and the design and adoption of agricultural[141] and conservation interventions[142,143]. Sullivan-Wiley et al.[143]. show how a human behavior-centered approach to designing conservation interventions explicitly considers the motivation, ability, and decision context of potential adopters and identifies the types of responses that can address each of these three spheres, along with specific intervention strategies for their implementation.

Our analysis has several limitations. The lack of a global NCS project database means we cannot assess the representativeness of our literature and survey samples. Thus, the external validity of our findings about leading constraint categories and individual constraints for specific pathways, countries or subregions is unknown. That said, the non-overlapping survey and literature samples, which likely have different sampling biases, share two of their respective top-three constraint categories and four of their respective top-seven constraints. This gives us some measure of confidence that our findings are indicative of the reality faced by NCS projects at large.

While sizeable, the evidence base we present is fairly heavily concentrated on a relatively small number of pathways and unequally distributed geographically. As a result, the information for many pathways in many subregions is limited, resulting in very small sample sizes for several pathway-subregion combinations. This is especially true for the regenerative agriculture pathway, which was excluded in Ref. 36. This incomplete spatial coverage of our dataset means that the absence of evidence of constraints in a particular location in our data should not be interpreted as evidence of the absence of constraints. Yet our subregion-level finding of shared constraint categories across

many pathways (Figure S17) suggests that many constraints reported for a specific pathway in a subregion likely also apply to other pathways in that subregion. Thus, it may be appropriate to infer the presence of a given constraint (or category) based on constraints (or categories) reported for other pathways in the subregion. Conversely, our subregional findings, which are based on the aggregation of documented constraints reported for country or subnational study and project areas, do not necessarily hold for all countries in a given subregion, let alone for all areas within a country. Interested readers should consult our constraints database for information about the specific study area covered by a paper or survey response. Future research on NCS should seek to systematically enlarge the evidence base presented here, focusing particularly on pathways and regions that are underrepresented relative to their mitigation potential.

The data on NCS constraints used in our analysis primarily represent assessments by academic researchers and NCS project staff. While often informed by local stakeholder input, these assessments nevertheless may not always accurately reflect local stakeholder perspectives.

Finally, we report the relative frequencies with which different constraints are reported. However, frequency of observation does not necessarily indicate importance, either in terms of a constraint by itself being sufficient to prevent NCS implementation, or of exacerbating or underlying other constraints. Assessing the near-term feasible NCS potential thus requires additional, geography-specific information about the causal relationships among prevalent constraints to identify the ultimate constraints in each geography, the near-term mutability of these constraints, effective interventions for overcoming or substantially mitigating these constraints, the cost of these interventions, and available financing. Similarly, a better understanding of the time scales over which individual constraints may be mutable and the relative difficulty and cost of mitigating key constraints in a particular geography would aid in prioritizing efforts to overcome constraints.

While our analysis documents that NCS projects often face a complex set of diverse types of constraints, it does not follow that widespread additional NCS implementation is infeasible. Indeed, the data we provide on documented barriers can help identify the actions needed to enable broader NCS adoption. This is true for both high-level efforts that seek to advance NCS implementation across entire countries or regions, and for individual NCS projects, the planning and design of which may benefit from our geo-referenced constraint information and systematic, comprehensive characterization of potential constraints that projects may want to consider at the scoping stage.

Nevertheless, our finding of many diverse challenges to NCS implementation suggests that absent concerted efforts to tackle constraints, the near-term feasible mitigation potential of NCS may be well below their documented biophysical potential. This underscores the argument that NCS are a complement to, not a substitute for drastic reductions in fossil fuel-based emissions[144,145], which, like properly-implemented NCS, produce their own large human wellbeing benefits[146–150].

Even where barriers to large-scale NCS implementation can be overcome in the near term, doing so will require resource investments. The size of those needed investments currently is unknown but not unknowable. Rather, it is determined by the sets of coordinated interventions that could remove the key, ultimate constraints on NCS in given geographies. Identifying what those key constraints and interventions are requires efforts that bring together potential NCS adopters, civil society, researchers and government.

Importantly, the justification for such investments does not lie only, or indeed even primarily, in the resulting climate change mitigation and reduction of climate change-related damages[57,151–153], but also in the multiple benefits for human well-being and biodiversity that NCS can deliver[92,94,154]. Crucially, many interventions required to remove constraints on NCS can be expected to generate human livelihood improvements unrelated to NCS. Examples of such interventions are increased agricultural and forestry extension services, improved tenure security and market, credit and insurance access and value chain integration of smallholders and disadvantaged communities, improved policy coordination, and reduced corruption. Consequently, such interventions should not be seen as having the sole or primary purpose of increasing NCS feasibility, but rather as being equally necessary for and justified by the broader livelihood and wellbeing gains they deliver to the affected populations.

While NCS have the potential to improve the livelihoods and wellbeing of local populations, these outcomes are not guaranteed. Indeed, the inappropriate or misguided design and implementation of NCS –and NbS more broadly – can have considerable negative consequences[1,155] especially for marginal, land-dependent communities[156]. This risk is particularly high for projects implemented through top-down governance structures that do not respect community rights, values, preferences and knowledge and that perpetuate power asymmetries[157]. It is thus imperative that NCS implementation follow best practice principles[7] to ensure equitable, socially inclusive designs that are community-driven and meet local preferences and needs[85], in addition to being sustainable, climate-additional, measurable, and nature-based. Adherence to these principles and to the conditions we identify for enabling broader NCS uptake may help overcome many of the observed constraints and further equitable, socially inclusive outcomes while mitigating climate change.

## Methods
The research presented here complies with all relevant ethical regulations. The survey was approved by The Nature Conservancy's (TNC) Chief Scientist under TNC's Human Subject Research Review. Free, prior and informed consent was obtained from all survey participants.

We combined a systematic literature review and a multilingual survey to identify implementation constraints on NCS (e.g.[158]). The rationale for this mixed-methods approach was threefold. First, the survey allowed us to probe more comprehensively for the presence of potential constraints at a study site, compared to published papers which often focus on one or a few specific constraints. The survey also allowed us to enlarge the peer-reviewed English language literature evidence base on NCS constraints, which shows uneven geographic and NCS pathway coverage[36], and may help counter the potential language and geographic biases in peer-review journals (e.g.[159]).

### Constraints typology
Following Brumberg et al.[36], we defined constraints as factors that negatively affect the adoption, viability, or effective implementation of NCS. We used mixed a priori and emergent coding to develop a list of implementation constraints. We extracted an initial list of constraints from a set of recent papers that identify constraints on specific NCS[23,32] or nature-based solutions for climate change mitigation[52,160]. We refined this initial list through a systematic review of recent peer-reviewed studies that document NCS constraints, which yielded 39 discrete constraints[36]. These were further refined through pre-testing of our NCS project survey instrument and analysis of survey responses. This further refinement resulted in three changes: a decomposition of several of Brumberg et al.'s[36] constraints into discrete component constraints, changes in the wording of some constraints to better encompass the diversity of specific permutations a constraint presented in the literature and survey, and the decomposition of Brumberg et al.'s 'economic' constraints category into 'finance' and 'markets' categories. Our final list contained 46 discrete constraints that we grouped into eight categories (Table 1): Material Inputs, Finance, Markets, Knowledge, Social-behavioral, Rules and Laws, Government and Organizations, and Negative Side Effects (see Supplementary Materials Table S1 for descriptions). Despite differences in terminology

and in some cases categorization, our list captures the constraints identified in other recent classifications[24–26,36] and often presents a finer parsing or decomposition of constraints (Supplementary Materials, Constraint terminology and categorization; Table S2). For example, Ref. 26. identifies high up-front costs as an economic constraint. However, the latter are more likely to be a barrier if access to credit is lacking (e.g.[161]) or if this up-front cost is not at least offset by increased profitability of NCS during a decision-relevant time period – two constraints that we capture separately. A crosswalk of our 46 constraints with those identified in other recent studies can be found in the Supplementary Materials (Constraint terminology and categorization; Table S2).

### Survey instrument and implementation

The survey was targeted at projects that implement activities that qualify as NCS (Figure S1), regardless of whether a project self-identified primarily by that activity (e.g., agroforestry) or implemented it as part of a broader portfolio of actions, and regardless of whether a project had stated climate mitigation goals. For the purposes of this survey, a 'project' was defined as (1) the direct implementation of NCS by an entity other than an individual landowner, or (2) the support of NCS implementation by others through the provision of technical expertise, supply of inputs, or some other activity. The survey was directed at project leads and their teams because a global survey of individual landowners was beyond the scope of this study. However, respondents were asked to report constraints faced by their project itself as well as by the land managers it engages. Project-level responses lead to the unavoidable filtering of individual land manager perceptions of constraints, but likely also promote a synthesis of constraints reported by individual land managers and thus facilitate the identification of the most important project-level constraints. The survey was implemented through an online questionnaire (Supplementary Materials, Survey questionnaire). Respondents were asked to provide information about select project characteristics (Supplementary Materials, Questionnaire development and distribution), identify the implemented NCS pathways and sub-pathways (Figure S1), and select from our list of 46 constraints all that made it challenging to achieve the full project objectives, with write-in options for additional constraints. The questionnaire was piloted with NCS projects in Brazil and Indonesia. To minimize the risk of misinterpretation, key terms in the questionnaire were identified through different font color, underlined, and accompanied by pop-up explanations. The questionnaire was originally developed in English and then professionally translated into French, Portuguese and Spanish, with translations checked by NCS practitioners who are native speakers in each language. The survey employed conditional (skip) logic and was implemented on the mobile-friendly Qualtrics platform. The survey was open from December 2022–December 2023, was distributed to prospective projects by the authors and colleagues at 16 conservation, development and research institutions, and was promoted in, and accessible via, the online newsletters of Restor (restor.eco) and Nature4Climate (https://nature4climate.org) (Supplementary Materials, Questionnaire development and distribution).

### Georeferencing of literature study sites and survey projects

All literature and survey projects were geo-referenced to their corresponding administrative (ADM) units at the ADM-0 (country), ADM-1 (state or equivalent), or ADM-2 (county or equivalent) level using the geoBoundaries dataset[162], based on the study area information provided in a paper and the implementation area reported for each surveyed project, respectively. If a paper or survey covered multiple ADM units and did not report to which ADM unit(s) an observed constraint applied, we assigned the constraint to all ADM units covered by the paper or survey. All data analysis was performed in R[163]. For additional information on geocoding and data analysis see Supplementary Materials.

### Included NCS pathways

Our analysis includes 22 NCS pathways as shown in Figure S1. These are commonly grouped into three NCS strategies – protection, improved management, and restoration (Figure S1[5]). The survey questionnaire referred to the increased soil carbon in grazing lands, reduced emissions in grazing lands, reduced emissions in croplands, and increased soil carbon in croplands pathways jointly as regenerative agriculture (other than agroforestry)[164], and our analysis maintains this grouping. Similarly, we also report the pooled results for the three agroforestry pathways (cropland-based agroforestry, silvopasture, forest-based agroforestry) and the two reforestation pathways (natural forest restoration, reforestation with plantations), respectively. The literature portion of our analysis is based on Brumberg et al.[36], which analyzed the constraints reported in recent (2020–2021) peer-reviewed papers that identify barriers to 13 NCS. Brumberg et al.'s analysis excluded regenerative agriculture pathways (other than agroforestry). We reviewed and recoded the studies included in Brumberg et al.[36] to the constraints used in this paper (Supplementary Materials, Recoding), and geo-referenced them to the corresponding ADM units. We screened the literature and survey datasets for overlap to avoid double-counting and found that no project was represented in both.

Because our analysis of constraints is implemented at the NCS pathway level, a survey project or paper that reported the same constraint for multiple sub-pathways of the same pathway – say, tree-based intercropping and silvopasture, two forms of agroforestry –generated a single constraint-pathway-location observation (hereafter: constraint observation). Conversely, a paper or survey that recorded the same constraint for two different pathways–say, agroforestry and reforestation–generated two constraint observations.

We count each unique instance of a constraint reported in a paper or survey response. For example, if a paper reported skepticism or disinterest in NCS among landowners, local government representatives, and national policymakers, we counted this as three counts of skepticism or disinterest because each instance independently affects NCS implementation feasibility. Consequently, the constraint counts for a pathway or subregion generally exceed the number of papers and survey responses for that pathway or subregion.

We present results primarily at the UN subregion level, which distinguishes 22 geographic subregions[165]. Because neither the literature nor the survey sample included two subregions (Micronesia, Polynesia), we present results only for the 20 subregions with data.

### Data analysis

In addition to the overall frequency counts of observations of individual constraints and constraint categories, we analyze observation counts by NCS pathway and UN subregion. We identify the most-frequently number one-ranking constraints and constraint categories for pathways and subregions, respectively, and the constraints and categories with the highest number of combined top-three rankings for pathways and subregions, respectively.

We assess the impact of different aggregation approaches for pathway and subregion-level data on these rankings. To illustrate how different aggregation approaches can impact constraint or constraint category rankings for a pathway or subregion, consider the case of aggregating pathway-level constraint observations in a subregion. The most straightforward way of doing so is to sum the observations of a given constraint for all pathways in the subregion. Under this raw count aggregation approach, the constraints that are most prevalent in better-studied pathways tend to emerge as the prevalent constraints for the subregion as a whole. This may lead to a biased assessment of

**Table 1 | Typology of constraints on NCS implementation**

| Category | Constraint |
|---|---|
| Material Inputs | Planting stock or other materials for the NCS |
| | Labor (external or own) for the NCS |
| | Suitable land for the NCS |
| | Water for the NCS |
| Finance | Land manager access to credit for NCS |
| | Land manager access to other funding for NCS |
| | Land manager insurance for NCS assets or outputs |
| | Project access to credit for NCS |
| | Project access to other funding for NCS |
| | Burdensome reporting requirements |
| Markets | Markets for NCS outputs produced by land managers |
| | Markets for carbon sequestered by the NCS |
| | Markets for ecosystem services or biodiversity provided by the NCS |
| | Prices for NCS outputs produced by land managers |
| | Prices for carbon sequestered by the NCS |
| | Prices for ecosystem services or biodiversity provided by the NCS |
| | Greater profitability of alternative land uses |
| Negative side effects of NCS | Negative side effects (resource use, human health, property damage, wildlife conflict) |
| Knowledge | Land manager literacy, numeracy, or technological capabilities |
| | Information about how to design or begin the NCS |
| | Information about how to manage the NCS |
| | Availability of technical advice for land managers |
| | Information about yields, inputs, or profits |
| | Information about market access or prices |
| | Information about on-site benefits of NCS |
| Social and Behavioral | Local preferences for non-NCS land uses |
| | Aversion to trying new land uses |
| | Skepticism or disinterest in NCS or lack of trust in NCS promoters |
| | Social norms favoring non-NCS land uses |
| | Concerns over negative equity impacts of NCS |
| | Lack of opportunity to participate in or influence the implementation of NCS due to gender, race, ethnicity, or other dimensions of identity |
| | Limited social learning or exchange networks for NCS |
| | Difficulty identifying, engaging, or coordinating with relevant actors |
| | Lack of dispute resolution |
| Rules and Laws | Insecure or uncertain rights to manage or sell property |
| | Insecure, uncertain, or lack of rights to use natural resources |
| | Regulatory barriers to production, transport, or sale of NCS outputs |
| | Insecure or uncertain NCS benefit sharing |
| | NCS-related corruption |
| | Unclear laws and policies related to NCS outputs/markets |
| | Financial or other incentives for non-NCS |
| Government and organizations | Lack of policy coordination or implementation capacity |
| | Uncertain, or lack of, enforcement of environmental laws |
| | Weak monitoring or enforcement of NCS agreements |
| | Violent conflict or perceived threat of violence |
| | Politically influential interests favoring non-NCS |

See Table S1 for descriptions

the prevalent NCS constraints in the subregion if the relative prevalence of constraints differs among pathways and the pathway composition of the evidence base does not reflect actual current or potential future pathway implementation extent. An alternative aggregation approach that overcomes this potential drawback assigns equal importance weights to all pathways, calculates for each pathway

a metric that indicates the relative prevalence of each constraint for the pathway, and then, for each constraint, sums the values of this metric over all pathways, to arrive at a subregional measure of relative prevalence that removes the impact of (i.e., normalizes for) differences in the pathway observation counts in that subregion. This approach allows constraints that are more prevalent in larger numbers of pathways in a subregion to emerge as the more prevalent constraints for the subregion as a whole, rather than the constraints that are more prevalent in better-studied pathways in the subregion. We implement this latter approach by calculating z-scores as a measure of relative prevalence of each constraint and constraint category for each pathway in a subregion and compare the results to the first approach that sums raw observation counts across pathways in each subregion. Because we found that the impact of the choice of aggregation approach on the rankings of constraints and constraint categories is marginal, we report the rankings based on pathway or subregion raw count aggregation (over subregions and pathways, respectively). However, we show how little results differ under the z-score aggregation approach, using the example of ranking the most-prevalent constraint categories in subregions.

To identify which constraints may be most important to address to unlock the global NCS mitigation potential, we also assess the constraint composition in each of the ten countries with the largest identified NCS biophysical mitigation potential, which together account for over half of the total identified global potential (3.24 of 5.85 Gt $CO_2$e yr$^{-1}$ [166]).

## Reporting summary
Further information on research design is available in the Nature Portfolio Reporting Summary linked to this article.

## Data availability
The geo-referenced database generated in this study has been deposited in the Harvard Dataverse Repository under accession code (https://doi.org/10.7910/DVN/EDZHBF).

## Code availability
The analysis code and the figure code generated in this study have been deposited in the Harvard Dataverse Repository under accession code (https://doi.org/10.7910/DVN/EDZHBF).

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

## Acknowledgements

We are grateful to the 215 individuals who took the time to complete our survey. We also thank the following individuals and institutions who helped distribute and promote the survey: Ademola Ajagbe, Saswati Bora, Paula Caballero, Mauricio Castro, Miguel Castro, Rane G. Cortez, Nicole DeMello, Mary Fales, Catherine Fitzgerald, Jan Glendening, Shi-teng Kang, Leonardo Lacerda, Sara Leavitt, Valerie Leung, Sara Mascola, William McGoldrick, Rachel Pasternack, Yves C. Paiz, Parker Raup, Camila Rodriguez, Sushil Saigal, Daniel Shemie, Rhita Simorangkir, Stephanie Simowski, Fernando Veiga, John Verdieck and Stephen Wood (The Nature Conservancy), Arthur Chinsman-Williams, David Tsetse and Olaf Westermann (Catholic Relief Services), Marta Zeymo (Conservation International), Seth Shames (EcoAgriculture Partners), Gena Gammie (Forest Trends), Mark Chandler (Heifer International), James Lloyd (Nature4Climate), John Mundy and Lori Pearson Von Coelln (One Acre Fund), Clara Rowe and Camellia Williams (Restor), Elizabeth Tully (Wildlife Conservation Society), Isabel Harrington (World Resources Institute), Jack Putz and Claudia Romero (University of Florida at Gainesville), Pham Thu Thuy (Center for International Forestry Research), Doris Mutta (African Forest Forum), Mani Nepal and Rajesh K. Rai (International Center for Integrated Mountain Development). This work was supported by a Bezos Earth Fund grant to The Nature Conservancy (T.K., J.T.E., Z.L., H.B., W.E., A.L., P.S., P.W.E.). H.B., M.H., and W.E. acknowledge support by the US Department of Agriculture (USDA) and the National Institute of

Food and Agriculture (NIFA) (Award number 2020-38420-30727). H.B. acknowledges additional support through a National Science Foundation Graduate Research Fellowship (Grant number DGE-2146755).

## Author contributions

T.K., J.T.E., P.W.E., and P.S. conceptualized the research. T.K., J.T.E., Z.L., H.B., A.L., W.E., M.H., P.S., L.E.O., D.M., P.H.S.B., M.B., A.J., K.G.A., A.T.K., M.S., and L.M. developed the methodology. T.K. and J.T.E. programmed the online survey. T.K., J.T.E., Z.L., H.B., A.L., W.E. and M.H. developed the analysis code and conducted the formal analyses. Z.L., J.T.E., and T.K. created the visualizations. T.K. wrote the original and final manuscript drafts. J.T.E., H.B., L.E.O., A.L., M.H., P.S., W.E., D.M., A.J., P.H.S.B., K.G.A., L.M., M.B., P.W.E., A.T.K., M.S., and M.E. reviewed the original draft. All authors approved the final manuscript. T.K. supervised the research and managed the project.

## Competing interests

P.H.S.B. is a partner at re.green, a forest restoration company. All other authors declare that they have no competing interests.

## Additional information

[1]Global Science, The Nature Conservancy, Arlington, VA, USA. [2]Department of Environmental Studies, Dartmouth College, Hanover, NH, USA. [3]Department of Environmental Studies, University of Colorado Boulder, Boulder, CO, USA. [4]Emmett Interdisciplinary Program in Environment and Resources, Stanford University, Stanford, CA, USA. [5]The Natural Capital Project, Stanford University, Stanford, CA, USA. [6]Department of Ecology and Evolutionary Biology, University of Colorado Boulder, Boulder, CO, USA. [7]Natural Climate Solutions Science, The Nature Conservancy, Portland, ME, USA. [8]Department of Earth System Science, Stanford University, Stanford, CA, USA. [9]Wildlife Conservation Society, New York, NY, USA. [10]World Resources Institute, Washington, DC, USA. [11]Department of Forest Sciences, University of São Paulo, São Paulo, Brazil. [12]Center for Carbon Research in Tropical Agriculture, University of São Paulo, São Paulo, Brazil. [13]The Center for International Forestry Research (CIFOR) and World Agroforestry (ICRAF), Nairobi, Kenya. [14]Nature Conservation Foundation, Mysore, Karnataka, India. [15]Conservation International, Arlington, VA, USA. [16]Eden: People+Planet, Nairobi, Kenya. [17]Catholic Relief Services, San Salvador, El Salvador. [18]Department of Geography, Autonomous University of Barcelona, Barcelona, Spain. [19]EcoDecision, Quito, Pichincha, Ecuador. [20]Present address: The Nature Conservancy, Durham, NC, USA. ✉e-mail: tkroeger@tnc.org

