## [Transparent Peer Review file · Nature Communications]

Global literature review and survey of implementation constraints on natural climate solutions

Corresponding Author: Dr Timm Kroeger

Version 0:

Reviewer comments:

Reviewer #1

(Remarks to the Author)

This is a thorough, detailed piece of work which contributes valuable insight into the failure of natural climate solutions to reach their full potential and identifies the principal constraints by region and pathway. Used well, this work could be highly impactful in allowing those concerned with NCS implementation to mitigate constraints and develop successful implementation strategies. The evidence presented supports the conclusions drawn; there is clear and honest enumeration of the limitations. There are no evident flaws in the data analysis. The methodology is clearly described, such that replicability is possible and meets a high standard.

Reviewer #2

(Remarks to the Author)

The manuscript presents a study that aims to identify, categorize, and map the barriers hindering the implementation of Natural Climate Solutions (NCS). The authors provide a comprehensive, spatially explicit assessment of NCS constraints worldwide by combining two complementary datasets: a systematic literature review and a global survey. Overall, I find the manuscript well written and methodologically sound, with only minor issues to address.

- Line 58: Insert "(AFOLU)" after "other land uses" so that readers immediately understand the acronym for Agriculture, Forestry, and Other Land Uses.
- Lines 849–850: The phrase "each were completed" is grammatically acceptable but stylistically awkward, as each is singular while were is plural. Consider revising to: "Each of the twenty-nine complete survey responses was completed by a group of respondents, representing a total of 215 individuals from 40 organizations, including environmental (38%) and development (18%) NGOs, research institutions (18%), governmental entities including utilities (10%), multi-sectoral entities (10%), and for-profit enterprises (8%)."
- Figure 1: Include the percentage that each category accounts for next to its corresponding bar to help readers quickly interpret the data.
- Figure S10: Adopt the same visual style as Figures S6 and S16 for consistency. Although the meaning can be inferred, please state in the caption what the red and green colors represent for clarity.

Reviewer #3

(Remarks to the Author)

The paper presents a valuable contribution to climate change mitigation and to the fields related to the protection, restoration, and sustainable management of nature through the implementation of appropriate solutions. The effort devoted to data collection, systematization, and synthesis, as well as to the interpretation of the results obtained from the analysis of the literature and survey outcomes, is clearly recognised and appreciated. The results are presented and discussed in a comprehensive way, both in the main text and in the supplementary materials. The paper also provides several insightful considerations on how to address and overcome the identified constraints, which adds significant value to the work.

However, I believe that the MS should better clarify the links with Brumberg et al. 2025 (<https://doi.org/10.1093/pnasnexus/pgaf173>). The MS combines both a survey and a review. My understanding is that the

review has been already (partially or entirely?) presented in Brumberg et al. 2025. If this is the case, the MS should be more explicit in describing what is really new (the survey + the integration between survey and review, I assume) from what has been already published. The introduction contains a reference to the published paper, but the results and methods are more ambiguous in reporting the outcomes of this study vs the ones of the previous study (e.g., some of the figures of the MS builds on figures from the published paper). It is also quite surprising that the introduction (e.g., in lines 75-76) does not mention the previous study in the relevant literature, considering that it contributed substantially to address the need mentioned in lines from 80 on.

Specific comments

1. Lines 112–119: I suggest moving this paragraph to the beginning of the “NCS pathway representation” section, so that a general overview of the results is provided before going into details about the most and least represented pathways.
2. Figure 1: Consider adding the total numbers at the end of each bar to improve readability (for instance, it is not immediately clear that government and organizations is the most frequently observed, as it visually appears similar to social-behavioral). Alternatively, you could adjust the x-axis interval (e.g., using 500 instead of 1000) or switch to percentages to facilitate comparison between the text and the figure. The same comment applies to Figure 2.
3. Line 157: As in Comment 1, a general overview of the results would be useful here before discussing the “most frequently reported” aspects.
4. Figure 2 and lines 159–162: Although Table S1 in the Supplementary Materials provides the full names of the constraints, it remains difficult to match them between the text and the figure, since the abbreviations and full names differ slightly (e.g., “lack of non-credit funding for land managers” vs. “land manager access to other (non-credit) funding for NCS”). I suggest adding serial numbers (in both the figure and Table S1) to make the correspondence clearer.
5. Line 210: The reference to figure S13 does not seem correct — please double-check it.
6. Figure S19: The image appears blurred and is difficult to read. Since abbreviations were used in Figure 2, consider adopting the same approach here to enlarge the text.
7. Lines 690–695: It seems that the third reason is missing in this list — please check.
8. Lines 702–705: This passage is somewhat difficult to understand — could you please provide an example to clarify your point?
9. Lines 733–734: Within brackets, you refer to “Supplementary Materials, Survey questionnaire and distribution,” but there is no section in the Supplementary Information with that exact title. I understand you may be referring to both the “Survey Questionnaire” and “Questionnaire development and distribution” sections. If so, please specify both names to make the reference clearer; otherwise, revise the reference according to your intended meaning.
10. Line 748: From this point onward, the text appears somewhat disorganised. I suggest adding a subheading instead of relying on line breaks.
11. Section starting at line 821 – “Sample frame”: This section already presents results — in fact, they seem to be the first results worth discussing to contextualise the others. I therefore recommend moving this section to the beginning of the Results, before “NCS pathway representation” (line 104).
12. Minor typos and formatting issues:
 - a. Line 68: Capitalise “implementation.”
 - b. Line 74: Missing spaces.
 - c. Line 690: Missing closing parenthesis “)”.
 - d. Line 716: There seems to be an extra or missing parenthesis.

Reviewer #4

(Remarks to the Author)

Version 1:

Reviewer comments:

Reviewer #3

(Remarks to the Author)

My previous remarks have been addressed in a satisfactory way and the MS has been improved and it is now ready for acceptance. No further comments.

Reviewer #4

(Remarks to the Author)

December 16, 2025

Dear referees,

Thank you for taking the time to review and provide thoughtful comments on our manuscript *'Implementation constraints on natural climate solutions: A global literature review and survey.'* We believe that we have addressed all comments and that the manuscript has gained in clarity as a result.

Please find below our responses to all referees' comments on the manuscript **NCOMMS-25-45305A**. The referee comments are shown verbatim followed by our responses in boldface.

Please note: all line numbers shown in our responses to the referees' comments refer to the revised manuscript with tracked changes.

Reviewer #1 (Remarks to the Author):

This is a thorough, detailed piece of work which contributes valuable insight into the failure of natural climate solutions to reach their full potential and identifies the principal constraints by region and pathway. Used well, this work could be highly impactful in allowing those concerned with NCS implementation to mitigate constraints and develop successful implementation strategies. The evidence presented supports the conclusions drawn; there is clear and honest enumeration of the limitations. There are no evident flaws in the data analysis. The methodology is clearly described, such that replicability is possible and meets a high standard.

Response: We appreciate your taking the time to review our manuscript. It is encouraging to hear that you think the work presented has the potential to be impactful.

Reviewer #2 (Remarks to the Author):

The manuscript presents a study that aims to identify, categorize, and map the barriers hindering the implementation of Natural Climate Solutions (NCS). The authors provide a comprehensive, spatially explicit assessment of NCS constraints worldwide by combining two complementary datasets: a systematic literature review and a global survey. Overall, I find the manuscript well written and methodologically sound, with only minor issues to address.

- Line 58: Insert "(AFOLU)" after "other land uses" so that readers immediately understand the acronym for Agriculture, Forestry, and Other Land Uses.

Response: Thank you. We have made this change (58).

- Lines 849–850: The phrase "each were completed" is grammatically acceptable but stylistically awkward, as each is singular while were is plural. Consider revising to: "Each of the twenty-nine complete survey responses was completed by a group of respondents, representing a total of

215 individuals from 40 organizations, including environmental (38%) and development (18%) NGOs, research institutions (18%), governmental entities including utilities (10%), multi-sectoral entities (10%), and for-profit enterprises (8%).”

Response: Thank you for this suggestion. However, the wording you suggest would change the meaning as it would state that we received 29 complete survey responses, which is incorrect. Rather, 29 of the complete survey responses we received were completed by groups of individuals rather than by a single individual. We mentioned this fact of several responses being completed by groups to explain why the total number of respondents (215) exceeds the total number of projects represented in our survey data (154).

To avoid the awkward phrasing in the original manuscript, we reworded the sentence as follows (139-142):

“In 29 cases, a group of individuals completed a survey for a project, resulting in a total of 215 survey participants from 40 organizations representing environment (38%) and development (18%) NGOs, research institutions (18%), governmental entities including utilities (10%), multi-sectoral entities (10%) and for-profit enterprises (8%).”

- Figure 1: Include the percentage that each category accounts for next to its corresponding bar to help readers quickly interpret the data.

Response: Thank you for this suggestion. We agree that showing the percentage of each category is helpful and have added it to each bar.

- Figure S10: Adopt the same visual style as Figures S6 and S16 for consistency. Although the meaning can be inferred, please state in the caption what the red and green colors represent for clarity.

Response: Thank you. We have changed the color coding for subregions and regions to the same applied in Figures S6 and S16. We also added the following text to the caption of Figure S10:

“Shading intensity is proportional to a subregion-pathway combination’s share in all constraint observations (red shading) or the subregional or regional shares in all constraints observations (green shading).”

For consistency, we also added similar text explaining the color coding to the caption of Figure S6:

“Shading intensity is proportional to a subregion-pathway combination’s share in all papers or survey projects (red shading) or the subregional or regional shares in all papers or projects (green shading).”

Reviewer #3 (Remarks to the Author):

The paper presents a valuable contribution to climate change mitigation and to the fields related to the protection, restoration, and sustainable management of nature through the implementation of appropriate solutions. The effort devoted to data collection, systematization, and synthesis, as well as to the interpretation of the results obtained from the analysis of the literature and survey outcomes, is clearly recognised and appreciated. The results are presented and discussed in a comprehensive way, both in the main text and in the supplementary materials. The paper also provides several insightful considerations on how to address and overcome the identified constraints, which adds significant value to the work.

However, I believe that the MS should better clarify the links with Brumberg et al. 2025 (<https://doi.org/10.1093/pnasnexus/pgaf173>). The MS combines both a survey and a review. My understanding is that the review has been already (partially or entirely?) presented in Brumberg et al. 2025. If this is the case, the MS should be more explicit in describing what is really new (the survey + the integration between survey and review, I assume) from what has been already published. The introduction contains a reference to the published paper, but the results and methods are more ambiguous in reporting the outcomes of this study vs the ones of the previous study (e.g., some of the figures of the MS builds on figures from the published paper). It is also quite surprising that the introduction (e.g., in lines 75-76) does not mention the previous study in the relevant literature, considering that it contributed substantially to address the need mentioned in lines from 80 on.

Response: Thank you for reviewing our manuscript carefully and for the very helpful suggestions. Thank you in particular for highlighting the need to better explain the specific contributions this manuscript makes to the identification and mapping of NCS implementation constraints, beyond Brumberg et al.'s (2025) work.

In addition to the global survey that expands the evidence base beyond the published literature, these contributions are as follows:

- **Disaggregation of several of the constraints used in Brumberg et al., which results in a larger set of constraints (46 compared to 39);**
- **Recoding of Brumberg et al.'s (2025) evidence base to the new, larger constraints set;**
- **Splitting Brumberg et al.'s "economic" constraint category into two categories, "finance" and "markets", to better reflect the distinct sets of challenges that finance and markets represent;**
- **Separating out NCS pathways that Brumberg et al. (2025) combined due to the small number of studies in the reviewed literature. Specifically, Brumberg et al. combined avoided coastal wetland conversion and coastal wetland restoration; avoided**

grassland conversion and grassland restoration; and avoided peatland conversion and peatland restoration, respectively. We consider these six pathways separately;

- Mapping of the spatial extents of the study areas in the papers analyzed in Brumberg et al. Brumberg et al. only recorded the country of a study (ADM0 level). In contrast, we extracted location information from the studies and mapped study locations to the relevant lower administrative levels (ADM1 or ADM2) that matched the study areas;
- Integration of the literature and survey datasets

We note that in the Supplementary Materials, our original submission already provided a detailed description of how the literature review constraints observations were recoded to the slightly larger set of constraints used in the present paper (see SM section ‘Recoding of Brumberg et al.’s constraints’).

In the main text methods section, we now more clearly describe the two-step process through which the list of constraints used in the present manuscript were derived. Specifically, we now state that we further refined the 39 constraints identified in Brumberg et al. through pre-testing of our survey and analysis of survey responses, resulting in the 46 constraints used in the present manuscript. (763-784)

We also added Brumberg et al. (2025) to the cited studies in line 778.

In the introduction, we changed the paragraph starting on line 86, which now reads as follows:

“Brumberg et al.’s³⁶ systematic review of the recent literature on NCS constraints was an important contribution to filling this critical knowledge gap. In this paper, we build on and advance that effort by conducting a global survey of NCS projects that uses an expanded set of implementation constraints, recoding Brumberg et al.’s³⁶ literature dataset to this expanded constraints set, georeferencing all survey and literature constraint observations to their corresponding administrative levels, and integrating the survey and literature datasets into a spatial database that reflects the current evidence base on NCS implementation constraints.”

We also added Brumberg et al. (2025) in line 100 as an additional reference (after reference #35) for national-level assessments and in line 102 as an example of studies that employed a coarser NCS pathway set (they combined coastal wetland restoration and avoided conversion pathways, the peatland restoration and avoided conversion pathways, and the grassland restoration and avoided conversion pathways, respectively).

Specific comments

1. Lines 112–119: I suggest moving this paragraph to the beginning of the “NCS pathway representation” section, so that a general overview of the results is provided before going into details about the most and least represented pathways. **Response: Thank you. We implemented this change.**

2. Figure 1: Consider adding the total numbers at the end of each bar to improve readability (for instance, it is not immediately clear that government and organizations is the most frequently observed, as it visually appears similar to social-behavioral). Alternatively, you could adjust the x-axis interval (e.g., using 500 instead of 1000) or switch to percentages to facilitate comparison between the text and the figure. The same comment applies to Figure 2. **Thank you. Reviewer #2 had a similar suggestion: to add to each bar the percentage each category accounts for in the total number of observations. We implemented that change. We also added an explanatory statement to the figure caption. We implemented the same changes to Figure 2.**

3. Line 157: As in Comment 1, a general overview of the results would be useful here before discussing the “most frequently reported” aspects.

Response: Thank you for this suggestion. We have moved the statements that present key high-level findings to the beginning of this section (206-216). These pertain to 1) the overall smooth frequency distribution of observation counts across constraints, with the exception of the most frequently reported constraint; and 2) constraints in all categories except Negative side effects are frequently encountered.

4. Figure 2 and lines 159–162: Although Table S1 in the Supplementary Materials provides the full names of the constraints, it remains difficult to match them between the text and the figure, since the abbreviations and full names differ slightly (e.g., “lack of non-credit funding for land managers” vs. “land manager access to other (non-credit) funding for NCS”). I suggest adding serial numbers (in both the figure and Table S1) to make the correspondence clearer.

Response: Thank you for catching this inconsistency in the wording of the “land manager access to other funding” constraint. We checked all instances of constraints in the manuscript and their wording now matches the full constraint names shown in Table S1. We believe this eliminates the possibility of confusion regarding which constraint we are referring to in a specific instance.

5. Line 210: The reference to figure S13 does not seem correct — please double-check it.

Response: Thank you for catching this error. We appreciate your careful review! We removed the reference to figure S13. This figure is correctly referenced elsewhere in the manuscript.

6. Figure S19: The image appears blurred and is difficult to read. Since abbreviations were used in Figure 2, consider adopting the same approach here to enlarge the text. **Response: Thank you for noticing this. We now use the abbreviated constraint names in this figure and have increased its resolution. We have implemented the same changes in Figure S21.**

7. Lines 690–695: It seems that the third reason is missing in this list — please check. **Thank you. The third reason is given in the last two lines in this paragraph (755-761): “...and may help counter the potential language and geographic biases in peer-review journals.”**

8. Lines 702–705: This passage is somewhat difficult to understand — could you please provide an example to clarify your point? **Thank you. An example of our decomposition of some constraints used in other classifications is provided in lines 779-782:** “For example, Ref. 26 identifies high up-front costs as an economic constraint. However, the latter are more likely to be a barrier if access to credit is lacking (e.g.,¹⁶²) or if this up-front cost is not at least offset by increased profitability of NCS during a decision-relevant time period – two constraints that we capture separately.”

9. Lines 733–734: Within brackets, you refer to “Supplementary Materials, Survey questionnaire and distribution,” but there is no section in the Supplementary Information with that exact title. I understand you may be referring to both the “Survey Questionnaire” and “Questionnaire development and distribution” sections. If so, please specify both names to make the reference clearer; otherwise, revise the reference according to your intended meaning. **Thank you for noticing this error. We were referring to the *Questionnaire development and distribution* section in the Supplementary Materials and now are referencing that section correctly (802).**

10. Line 748: From this point onward, the text appears somewhat disorganised. I suggest adding a subheading instead of relying on line breaks. **Thank you for this suggestion. We have added two new subheadings, *Georeferencing of literature study sites* (817) and *survey projects and Included NCS pathways* (825).**

11. Section starting at line 821 – “Sample frame”: This section already presents results — in fact, they seem to be the first results worth discussing to contextualise the others. I therefore recommend moving this section to the beginning of the Results, before “NCS pathway representation” (line 104). **Thank you for this suggestion. We moved the *Sample frame* section to the beginning of the Results section (111-142).**

12. Minor typos and formatting issues:

a. Line 68: Capitalise “implementation.” **Corrected, thank you (68).**

b. Line 74: Missing spaces. **Space added, thank you (74).**

c. Line 690: Missing closing parenthesis “)”. **Added, thank you (756).**

d. Line 716: There seems to be an extra or missing parenthesis. **Thank you. We added the missing opening parenthesis (783).**

Reviewer #4 (Remarks to the Author):

Sincerely,

Timm Kroeger, Ph.D. (on behalf of all co-authors)
Senior Environmental Economist
Global Science
The Nature Conservancy
6636 Eames Way
Bethesda, MD 20817, USA
Email: tkroeger@tnc.org
Ph: (301) 728-7776